Urban forests sustain diverse carrion beetle assemblages in the New York City metropolitan area

Fusco Nicole A. 1
Zhao Anthony 2
Munshi-South Jason jason@nycevolution.org jmunshisouth@fordham.edu 1
1 Louis Calder Center–Biological Field Station, Fordham University , Armonk , NY , USA
2 Department of Entomology, University of Maryland at College Park , College Park , MD , USA
Sanders Nathan
Electronic publication date: 2017 Mar 15
Publication date: 2017
Volume: 5
Electronic Location ID: e3088
Received 2016 Oct 3; Accepted 2017 Feb 14
Copyright: ©2017 Fusco et al.
Copyright year: 2017
Copyright holder: Fusco et al.
License: This is an open access article distributed under the terms of the Creative Commons Attribution License, which permits unrestricted use, distribution, reproduction and adaptation in any medium and for any purpose provided that it is properly attributed. For attribution, the original author(s), title, publication source (PeerJ) and either DOI or URL of the article must be cited.
License URL: https://creativecommons.org/licenses/by/4.0/

Keywords: Carrion beetles, New York City, Urban parks, Urbanization, Urban ecology, Silphidae

Funding: National Science Foundation REU Site 1063076 Fordham University and the Louis Calder Center A Zhao’s work on this project was supported in part by a National Science Foundation REU Site grant (No. 1063076) to Fordham University and the Louis Calder Center. No additional external funding was received for this study. The funders had no role in study design, data collection and analysis, decision to publish, or preparation of the manuscript.

==============================
Urbanization is an increasingly pervasive form of land transformation that reduces biodiversity of many taxonomic groups. Beetles exhibit a broad range of responses to urbanization, likely due to the high functional diversity in this order. Carrion beetles (Order: Coleoptera, Family: Silphidae) provide an important ecosystem service by promoting decomposition of small-bodied carcasses, and have previously been found to decline due to forest fragmentation caused by urbanization. However, New York City (NYC) and many other cities have fairly large continuous forest patches that support dense populations of small mammals, and thus may harbor relatively robust carrion beetle communities in city parks. In this study, we investigated carrion beetle community composition, abundance and diversity in forest patches along an urban-to-rural gradient spanning the urban core (Central Park, NYC) to outlying rural areas. We conducted an additional study comparing the current carrion beetle community at a single suburban site in Westchester County, NY that was intensively surveyed in the early 1970’s. We collected a total of 2,170 carrion beetles from eight species at 13 sites along this gradient. We report little to no effect of urbanization on carrion beetle diversity, although two species were not detected in any urban parks. Nicrophorus tomentosus was the most abundant species at all sites and seemed to dominate the urban communities, potentially due to its generalist habits and shallower burying depth compared to the other beetles surveyed. Variation between species body size, habitat specialization, and % forest area surrounding the surveyed sites also did not influence carrion beetle communities. Lastly, we found few significant differences in relative abundance of 10 different carrion beetle species between 1974 and 2015 at a single site in Westchester County, NY, although two of the rare species in the early 1970’s were not detected in 2015. These results indicate that NYC’s forested parks have the potential to sustain carrion beetle communities and the ecosystem services they provide.

Introduction

The ecological influence of urbanization is increasingly pervasive around the world. In 2014, 54% of the world’s human population resided in urban areas (United Nations, 2014) and urban populations increased by 12% between 2000 and 2010 in the United States (United States Census Bureau, 2010). Urban landscapes are highly modified for human use, with natural habitats typically occurring only in small, fragmented patches (Saunders, Hobbs & Margules, 1991). Fragmentation in cities often decreases species richness, changes community composition, and alters ecosystem processes (Didham, 2010). Many species are impacted negatively by urbanization (carnivores—Ordeñana et al., 2010; arthropods—Sattler et al., 2010; amphibians—Hamer & Parris, 2011; birds & plants—Aronson et al., 2014), but effects vary based on the taxonomic group in question (McKinney, 2008). For example, studies of arthropod diversity along urban-to-rural gradients have documented a wide variety of responses to urbanization (Hornung et al., 2007; Niemelä & Kotze, 2009; Varet, Pétillon & Burel, 2011; Magura, Nagy & Tothmeresz, 2013; Savage et al., 2015; Diamond et al., 2015). Given the extreme variety of life history traits and habitat use among arthropods, responses to urbanization may be difficult to predict.

Carrion beetles use small mammal carcasses as food sources for their young (Scott, 1998). These beetles bury carcasses to avoid competition with other scavengers, thus facilitating decomposition and providing considerable ecosystem services. Availability of carrion likely influences the abundance and diversity of carrion beetles. Carrion beetle species compete with each other as well as with other scavenging vertebrates (Scott, 1998; Trumbo & Bloch, 2000; DeVault et al., 2011), and invertebrates for this resource (Ratcliffe, 1996; Gibbs & Stanton, 2001). Urbanization alters natural habitats in myriad ways (Grimm et al., 2008) that may cause local extirpations or reduced abundance of native small mammals (e.g., likely carrion) and carrion beetles in cities.

Gibbs & Stanton (2001) previously reported that forest fragmentation reduced carrion beetle species richness and abundance in Syracuse, New York. Beetles that persisted in these fragments were primarily small-bodied habitat generalists, and other carrion beetles may have declined in abundance due to lower carcass availability, increased prevalence of other scavengers, or reduced soil and litter quality. Wolf & Gibbs (2004) also found that forest fragmentation decreased carrion beetle diversity and abundance in Baltimore, Maryland. They argued that forest contiguity was an important factor affecting richness, abundance, and diversity of carrion beetles in this city. However, these studies did not directly address whether large parks within core urban areas harbor a substantial diversity of carrion beetles.

In this study we investigated species richness, diversity, relative abundance and community similarity of carrion beetles (Family: Silphidae) across an urban-to-rural gradient in the New York City (NYC) metropolitan area. NYC is the most densely populated area in North America, but 27% of the city’s land area is comprised of vegetated natural areas, particularly within several large urban parks (New York City Department of City Planning, 2002). These parks are characterized by substantial forest cover and high densities of small mammals (Munshi-South & Kharchenko, 2010), and thus may provide high-quality habitat for a diverse assemblage of carrion beetles. Alternatively, NYC’s urban forests may harbor less carrion beetle diversity relative to suburban and rural areas outside of NYC as was found in Baltimore and Syracuse (Gibbs & Stanton, 2001; Wolf & Gibbs, 2004). We also compared historical records from a single site (Pirone & Sullivan, 1980), the Louis Calder Center in Armonk, New York, with our 2015 estimates of carrion beetle diversity and abundance to examine changes over the last four decades. The forest area at the Louis Calder Center has not changed in that time, but urbanization of the surrounding area and a rapid increase in deer herbivory may have resulted in altered diversity of community composition of carrion beetles.

Urban forests have the potential to harbor substantial biodiversity in cities worldwide (Faeth, Bang & Saari, 2011; Elmqvist et al., 2013; La Sorte et al., 2014; Aronson et al., 2014). Diamond et al. (2015) argued that increases in biodiversity may be due to introduction of non-native species as well as increased habitat heterogeneity in densely populated areas, but many native species still persist in these urban remnants. Urban carrion beetle diversity that rivals surrounding rural areas would indicate that urban forests in NYC currently provide ample habitat and resources to sustain these native beetle communities.

Table 1 Characteristics, classification and quantification of urbanization at each sample sites, along with species richness, and species diversity measures at each site (urban = orange, suburban = teal, rural = purple).

Site code	Site name	Site classification	Mean % impervious surface	Species Richness (species number)	Species Diversity (Simpson’s 1/D)	
NYBG	New York Botanical Garden	Urban	60.88%	2	1.61	
HBP	High Bridge Park	Urban	60.54%	6	1.84	
CP	Central Park	Urban	60.24%	6	2.66	
IHP	Inwood Hill Park	Urban	29.97%	6	2.94	
VCP	Van Cortlandt Park	Urban	27.97%	5	2.34	
SWP	Saxon Woods Park	Suburban	17.63%	5	2.37	
LCC	Louis Calder Center	Suburban	10.57%	–	–	
LCC1	Louis Calder Center Sample 1	–	–	3	3.20	
LCC2	Louis Calder Center Sample 2	–	–	4	2.59	
LCC3	Louis Calder Center Sample 3	–	–	5	2.87	
CSH	Convent of Sacred Heart	Suburban	11.14%	4	2.31	
RSP	Rockefeller State Park	Suburban	4.74%	5	1.75	
MRG	Mianus River Gorge Preserve	Suburban	0.64%	6	2.59	
CT	Western Connecticut	Rural	0.89%	3	2.38	
CAT	Catskills	Rural	0.46%	5	1.68	
CFP	Clarence Fahnestock State Park	Rural	0.20%	4	1.77	

Materials & Methods

Study site and sampling techniques

This study was conducted across an urban-to-rural gradient spanning 120 km from the urban core of NYC (Central Park, Manhattan) to southern New York State and western Connecticut. Carrion beetles were collected from five urban sites in New York City, five suburban sites and three rural sites (Table 1 and Fig. 1). Urban, suburban and rural site classifications followed Munshi-South, Zolnik & Harris (2016) and were based on percent impervious surface cover. Many of our sampling sites were previously used by Munshi-South, Zolnik & Harris (2016) to examine population genomics of white-footed mice (Peromyscus leucopus) and thus classifications were already available. For sites unique to this study, we used the same methods to quantify urbanization (Table 1 and Fig. 1). In brief, we created 2 km boundary buffers around our study sites in ArcGIS 10.3 (ESRI, 2014) and then used zonal statistics to calculate mean percent impervious surface from the Percent Developed Imperviousness data layer imported from the National Land Cover Database 2011 (Xian et al., 2011).

Figure 1 Geographic location of study sites surrounded by 2 km buffer circles. Urban (orange), suburban (teal), and rural (purple) sites were classified according to impervious surface in these buffers as described in the text.

Green areas represent no impervious surface, whereas areas of increasing pink coloration denote increasing percent impervious surface as reported in the 2011 National Landcover Database (Xian et al., 2011). Site abbreviations follow Table 1.

For comparison with previous carrion beetle surveys in other cities, we followed the sampling methods and trap design employed by Gibbs & Stanton (2001) and Wolf & Gibbs (2004). We constructed traps from open-topped cylinders by cutting the top off 1 L plastic bottles, adding a loop of string to hang the trap, and attaching a rain cover (cardboard covered in plastic cling wrap) by threading it through the string. Most traps contained 200 mL of a 1:1 mixture of ethylene glycol and water, although soapy water was substituted at the Manhattan sites due to public safety regulations. A small glass jar containing bait (∼6.5 cm2 of rotting chicken thigh) was topped with a punctured lid to prevent insects from destroying the bait but permitting odors to attract carrion beetles. This jar was placed inside each plastic trap that was then filled with the ethylene glycol mixture. We set three traps at each site, close to forest edges and at least 100 m apart. Traps were hung from small tree branches approximately 1–1.5 m from the ground to prevent other wildlife from disturbing the traps. Traps were set out for seven consecutive days at each site, where beetles were collected upon the last day. We conducted all trapping from 22 June to 05 August 2015. At each study location, we separated beetles from other insects, and stored beetles in 80% ethanol before bringing all specimens to the laboratory for identification. We identified all carrion beetle species following Hanley & Cuthrell (2008). After collection, all beetles were stored in ethanol at −20 °C. Permission to collect carrion beetles was granted by the New York City Department of Parks and Recreation, the Rockefeller State Park Preserve, and the Connecticut Department of Energy and Environmental Protection (Permit number: 1214008).

Relative abundance at urban, suburban and rural sites

To describe variation among sites and site classes we calculated relative abundance as the proportion of each species compared to the total number of individuals at each site. To examine changes over the summer season in carrion beetle diversity and relative abundance at the Louis Calder Center (a suburban site), we conducted three separate trapping surveys, from 22–29 June, 14–21 July, and 22–29 July 2015. Only data collected from the third survey were used in the main urban-to-rural analysis to more accurately compare to samples taken during the same time period as sampling at the other sites (mid to late July 2016). We calculated the relative proportion of each species for each of the three surveys at the Louis Calder Center to examine changes throughout the study period.

Historical comparison of carrion beetle presence and abundance

Pirone & Sullivan (1980) performed carrion beetle sampling at the Louis Calder Center in Armonk, NY for an 8-month period (April–November) in 1974. They collected 4,300 silphid beetles in 6 pitfall traps. Although our current study is only a snapshot (3 weeks of sampling) of the current community assembly at this site, the current data collected in 2015 from all three surveys at the Louis Calder Center (June–July) were used to compare the current carrion beetle species (2015) with the species observed in the historical study (1974). A student’s t-test was conducted to compare total relative abundance in 1974–2015.

Species diversity and species richness along an urban-to-rural gradient

To determine whether our sample size was robust enough for running subsequent statistical analyses we performed a rarefaction analysis for all sites in R v.3.2.3 (R Core Team, 2015) using the vegan package. For the analysis along the urban-to-rural gradient we used two different measures; species richness and species diversity. To compare species diversity across sites, we calculated the Simpson’s reciprocal index (1/D; Simpson, 1949) of diversity. We also calculated other diversity indices for comparison (equations; Jost, 2006; Table S1). These results showed similar trends across sites for all indices, therefore we chose to use the very commonly used Simpson’s reciprocal index for statistical analyses. We then calculated community similarity using the Jaccard Index of community similarity (Jaccard, 1901): CCJ=SJ=a∕a+b+c,

where SJ is the Jaccard similarity coefficient, a is the number of species shared by all sites, b is the number of species unique to the first site, and c is the number of species unique to the second site. Then we calculated the Jaccard coefficient of community similarity for all the data pooled across sites classes (urban, suburban and rural) to analyze overall carrion beetle community assemblages. Lastly, we conducted a hierarchical cluster analysis using the betapart package in R (Baselga & Orme, 2012) to explore patterns of beta diversity partitioning this diversity measure into the nestedness and the turnover components.

To examine the influence of urbanization on species richness and diversity, we calculated general linear regressions of mean percent impervious cover versus species richness and species diversity at each site using R. We also calculated a general linear regression to explore the difference in relative abundance of the most abundant species; Nicrophorus tomentosus versus mean percent impervious surface at all sites. Additionally, we performed a gradient analysis by creating a Non-metric Multidimensional Scaling (NMDS) plot using the Vegan package (Oksanen et al., 2016) in R to investigate population dissimilarity based on site class (urban, suburban, and rural). NMDS compares species changes from one community to the next by using rank order comparison and calculates the pairwise dissimilarity of points in low-dimensional space (Buttigieg & Ramette, 2014). Thus, NMDS allows us to robustly estimate dissimilarity between site locations based on the type of site and the species located in each site.

Species-specific differences across urban, suburban, and rural sites

Many studies on beetles have focused on specific characteristics that may underlie differences in species richness and diversity within sites and across studies (Davies & Margules, 2000). We conducted a factorial ANOVA to examine the interaction effect of species body size (small < 5 mm, medium = 5–6.5 mm, large > 6.5 mm; estimations and groupings based on data from Gibbs & Stanton, 2001) and site classification (urban suburban, rural) based on relative abundance. We also used a Student’s t-test to examine relative abundance when species are classified as habitat generalists versus habitat specialists (Gibbs & Stanton, 2001). Lastly, we performed a general linear regression to explore if a relationship exists between species richness or species diversity and the forest area (%) existing at each site. At each site the forest area was calculated using the Tabulate Area tool from the ArcGIS 10.3 (ESRI, 2014) Toolbox to calculate the forest area within the same 2 km buffers surrounding each site as was used to calculate percent impervious surface. We then calculated the relative proportion of forest area compared to the total area within the buffer. We used forest area data from the USGS National Landcover Dataset (Homer et al., 2011).

Results

Relative abundance at urban, suburban and rural sites

We collected a total of 2,170 carrion beetles comprising eight silphid species (Table 2) across all sites (Table 1, Fig. 1). Nicrophorus tomentosus was the most abundant at all sites, accounting for 56.8% of all beetles captured (Fig. 2), yet there was no significant relationship between Ni. tomentosus relative abundance and percent mean impervious surface of each site (F(2, 10) = 1.16, p > 0.05). Other species also varied in presence or abundance between urban, suburban and rural forests (Fig. 2); specifically, Oiceoptoma noveboracense was captured more often in suburban areas (23.5%) and urban areas (21.9%) than at rural sites (1.6%), whereas Nicrophorus defodiens was captured predominantly in rural areas (7.1%) versus urban areas (0.3%) and suburban areas (0.2%). Similarly, Necrophila americana was recorded in suburban (8.4%) and rural areas (4.3%) but was not found in any urban sites. Nicrophorus sayi was only recorded at one rural park, accounting for 1.4% of the total number of beetles captured at rural sites (Tables 1 and 2, and Fig. 2).

Table 2 Abundance data for eight carrion beetle (Family: Silphidae) species at all site locations (site abbreviations and classification found in Table 1).

Species	Nicrophorus orbicollis	Nicrophorus tomentosus	Nicrophorus defodiens	Nicrophorus pustulatus	Nicrophorus sayi	Oiceoptoma inaequale	Oiceoptoma noveboracense	Necrophila americana	
									
NYBG	50	165	1	8	0	25	77	0	
HBP	3	28	0	0	0	1	7	0	
CP	0	6	0	0	0	0	19	0	
IHP	61	70	1	1	0	1	20	0	
VCP	22	75	0	1	0	3	75	0	
SWP	7	87	0	15	0	1	125	0	
LCC	39	257	0	3	0	0	55	41	
LCC1	8	54	0	0	0	0	28	9	
LCC2	26	168	0	1	0	0	7	25	
LCC3	5	35	0	2	0	0	20	7	
CSH	50	182	2	0	0	0	11	53	
RSP	17	90	0	0	0	0	74	0	
MRG	26	221	35	0	0	0	6	4	
CT	1	97	0	0	0	0	15	20	
CAT	63	106	0	5	7	0	2	17	
CFP	50	165	1	8	0	25	77	0	

Figure 2 Relative abundance (%) of species across site classes; urban (orange), suburban (teal), rural (purple) sites.

Bold lines within the boxes indicate the median value, the colored boxes represent the inter-quartile range (Quartile 1–Quartile 3), the whiskers extend 1.5 * IQR, and the dots represent outlier values.

Figure 3 Relative abundance (%) of species across three sampling surveys (Late June, Mid July, Late July) at the Louis Calder Center site in Armonk, NY.

Historical comparison of carrion beetle presence and abundance

We captured 358 individuals cumulatively across three surveys at the Louis Calder Center throughout summer 2015. Five carrion beetle species were observed at Louis Calder Center with the absence of Ni. defodiens, Ni. sayi and Oiceoptoma inaequale found at other suburban and rural sites. Nicrophorus pustulatus and Necrophila americana were absent from the first surveys and appeared in later surveys. Ni. tomentosus became more prevalent (34.4%, 54.5%, 74.0%) throughout the summer and O. noveboracense decreased in relative abundance (34.4%, 28.3%, 3.1%; Fig. 3).

Results of carrion beetle observations at the Louis Calder Center site show a small reduction in species richness; 7 species in 1974 to 5 species in 2015 (Fig. 3). Ni. defodiens, Ni. sayi, and O. inaequale were absent both historically and currently at this site yet were present at other suburban sites. The two species not observed in 2015 that were already low in relative abundance in 1974 were Necrodes surinamensis (0.2%) and Necrophilus pettiti (0.3%; Fig. 4). There was no significant difference in the relative abundance of species in 1974 to 2015 (t(9) = 0.546, p = 0.599). The 2015 data show an increasing trend in the relative abundance of Ni. tomentosus (5.8%–54.3%) in 1974 versus 2015, and a decreasing trend in the relative abundance of Necrophila americana (41.8%–6.7%) and O. noveboracense (38%–21.9%) since 1974 (Fig. 4).

Figure 4 Relative abundance of species at the Louis Calder Center site in 1974 (Pirone & Sullivan, 1980) and in 2015.

Species diversity and species richness along an urban-to-rural gradient

Across the urban-to-rural gradient there was no significant relationship between mean percent impervious surface of a site and carrion beetle species richness (R2 = 0.028, p > 0.05) or species diversity (R2 = 0.0213, p > 0.05). However, NMDS ordination plots exhibit dissimilarity in carrion beetle assemblages in rural and urban sites. NMDS also showed that beetle assemblages in suburban sites were more similar to those in the urban sites. Based on the size of the convex hulls, heterogeneity of carrion beetle species composition was the greatest for suburban sites and least for urban sites (Fig. 5).

We observed relatively high community similarity indices across most pairwise comparisons (CCj = 0.333–1.000; Table 3). Several pairs of nearby sites had very high community similarity, such as two urban sites: Inwood Hill Park and New York Botanical Garden (CCj = 1.000); and two suburban sites: Convent of the Sacred Heart School and Louis Calder Center (CCj = 1.000). The most distant pairs of sites were less similar, most notably between highly urbanized Central Park and rural Clarence Fahnestock State Park (CCj = 0.333). The pooled urban carrion beetle community was more similar to the pooled suburban community (CCj = 0.857) than to the pooled rural community (CCj = 0.750), as also demonstrated in the NMDS ordination (Fig. 5). The pooled rural community was equally similar to both the pooled urban and suburban communities (CCj = 0.750). The nestedness component of beta diversity in the hierarchical cluster analysis clustered sites based on species richness trends, not based on site classifications (Fig. 6A). Alternatively, the hierarchical clustering of the turnover component clusters all urban sites together with a few suburban sites in one branch and all rural sites and the other suburban sites in another cluster (Fig. 6B) as reflected by the results of the NMDS plot (Fig. 5).

Figure 5 Non-metric multidimensional scaling (NMDS) of abundance of carrion beetle species at each site grouped by site class as a convex hull.

Table 3 Pairwise Jaccard community similarity index values calculated between all sample sites (site abbreviations and classification located in Table 1).

	NYBG	HBP	CP	IHP	VCP	SWP	LCC	CSH	RSP	MRG	CT	CAT	CFP	
NYBG	–	0.667	0.333	1.000	0.833	0.833	0.571	0.571	0.571	0.500	0.571	0.625	0.500	
HBP		–	0.500	0.667	0.800	0.800	0.500	0.500	0.500	0.750	0.500	0.750	0.429	
CP			–	0.333	0.400	0.400	0.400	0.400	0.400	0.667	0.400	0.500	0.333	
IHP				–	0.833	0.833	0.571	0.571	0.571	0.500	0.571	0.750	0.500	
VCP					–	1.000	0.667	0.667	0.429	0.600	0.429	0.875	0.571	
SWP						–	0.667	0.667	0.429	0.600	0.429	0.875	0.571	
LCC							–	1.000	0.667	0.600	0.667	0.875	0.833	
CSH								–	0.667	0.600	0.667	0.571	0.833	
RSP									–	0.600	1.000	0.625	0.833	
MRG										–	0.600	0.875	0.500	
CT											–	0.875	0.600	
CAT												–	0.875	
CFP													–	

Figure 6 Hierarchical cluster analysis of sites (abbreviations found in Table 1) based on (A) the turnover component, and (B) the nestedness component of the Jaccard Similarity Index for beta diversity.

Species-specific differences across urban, suburban, and rural sites

We found no significant effect between beetle body size and site class (urban, suburban, rural) for relative abundance of carrion beetle species in this study. Additionally, when focusing on habitat specialization, we also found no significant difference in habitat specialization between urban, suburban, and rural sites (Table S1). Lastly, we found no trend in species richness (Fig. S1A) or species diversity (Fig. S1B) across increasing continuous forest areas throughout sampled sites.

Discussion

Contrary to our predictions, we observed few differences in beetle diversity or richness along an urban-to-rural gradient in the NYC metropolitan area. Along this gradient, urban and suburban sites were nearly equally diverse and species-rich as rural sites. We also detected little to no influence of urbanization (measured by mean percent impervious surface) on relative species abundance, species diversity or species richness of carrion beetles throughout these sites. Although species richness is not very high, we observed relatively high carrion beetle community similarity values between most pairs of sites in this study regardless of their urbanization status (Jaccard Index = 0.333–1.000; Table 3). When partitioning beta diversity, nestedness follows species richness trends, further strengthening the result that there is no difference in carrion beetle communities between urban, suburban and rural classified sites. Alternatively, turnover may be driven by site class based on urbanization to some extent.

Overall, even with limited sampling, this study demonstrates that a diverse community of carrion beetles are able to thrive in rural, suburban and urban forests in and around New York City. Gibbs & Stanton (2001) and Wolf & Gibbs (2004) reported that carrion beetle diversity is significantly reduced around Syracuse, NY and Baltimore, MD due to forest fragmentation associated with urbanization, but our results indicate that forested city parks in the most urbanized areas of North America (i.e., Manhattan and the Bronx, NYC) do harbor substantial carrion beetle diversity compared to surrounding rural areas. We did not directly examine variation in fragment size within urban, suburban, and rural areas, but the discrepancy between these earlier results and ours may be due to the fact that urban forests in NYC parks are relatively large compared to other cities.

Arthropods have exhibited highly variable, even dichotomous, responses to urbanization around the world (Lessard & Buddle, 2005; Sattler et al., 2010). Other studies show that the influence of urbanization varies based on taxonomic group, geographic location, climate and spatial scale (McKinney, 2008; Kotze et al., 2011; Martinson & Raupp, 2013). Carabid beetles are the most well-studied arthropod group: carabid species richness has been found to decrease (Gaublomme et al., 2008) or not change (Deichsel, 2006) in urban areas. In a review of carabids and urbanization, Magura, Lovei & Tothmeresz (2010) argued that this variation was due to site-specific effects operating in each study. Urbanization did not homogenize carabid assemblages in cites in England, Denmark, and Helsinki, Finland, but urbanization did affect species assemblages in other Finnish cities, Hungary, Japan and Bulgaria (Kotze et al., 2011). Other than location, discrepancies across studies may be related to the choice of variables for analysis. Differing climates, different measures of urbanization (e.g., human population density, economics, housing density, or impervious surface), different times of the year studies were conducted, and spatial scale (Faeth, Bang & Saari, 2011) all could affect species presence, richness, and diversity results in urban areas. Although carrion beetles may not be representative of all arthropod species, these same factors may also explain some of the differences between the NYC results presented here, and those of Gibbs & Stanton (2001) and Wolf & Gibbs (2004) for Syracuse, NY and Baltimore, MD.

When quantifying and classifying urbanization, many studies use measures based on forest fragmentation, extent of forest cover, isolation caused by human-induced disturbance, impervious surface, or human population density (McDonnell & Hahs, 2008). The use of a common index to represent urbanization is necessary to compare results across studies, but there are no common indices currently in wide use. McDonnell & Hahs (2008) and Kotze et al. (2011) stress the need for such common measures to examine the generality of the influence of urbanization on biodiversity patterns. Comparing our study to other carrion beetle studies, we used mean percent impervious surface to quantify urbanization, whereas Gibbs & Stanton (2001), Wolf & Gibbs (2004) and Klein (1989) utilized continuous forest cover and fragmentation to classify the level of urbanization at each site. Our study specifically quantified urbanization with mean percent impervious surface using 2 km buffers as was previously reported by Munshi-South, Zolnik & Harris (2016) for many of the same study sites. Although many metrics can be employed to measure urbanization, we believe that impervious surface cover is particularly useful for urban-to-rural gradient studies because it is measurable for nearly any terrestrial area and directly related to urban landscape modification.

We also examined species relative abundance changes over three sampling periods at a suburban site: the Louis Calder Center in Westchester County, NY. Despite no significant change in total abundance across samples throughout the summer season, we did see slight changes in species abundance over different collection periods (Fig. 3). In comparison to Wolf & Gibbs (2004) our study showed opposite trends in species presence across the collection period. This difference could be due to difference in climate and temperature across June, July, and August in NYC versus the more southern Baltimore, MD. In Armonk, NY, Ni. tomentosus, O. noveboracense, and O. inaequale were more abundant later in the summer, and Ni. orbicollis was more abundant earlier in the summer. Scott (1998) describes Ni. tomentosus as a late summer / early fall breeder, which was corroborated by our observation that Ni. tomentosus was most abundant in the latest summer survey. Ni. orbicollis begins breeding in late spring (Ratcliffe, 1996), and was the most abundant in our first sample session at this site. Ni. sayi is most active in very early spring, which may explain the absence of this species at this site during the summer. Lastly, the absence of Ni. pustulatus may be due to their habitat preference for wetlands (Gibbs & Stanton, 2001), which were not very extensive around our trap sites in this study.

Species richness of carrion beetles did not differ greatly between 1974 and 2015 at the Louis Calder Center site. However, we did record pronounced species-specific differences in abundance between the past study (Pirone & Sullivan, 1980) and this current study in 2015 (Fig. 4). Species differences over time could be due to anthropogenic modification of the landscape in and around the Louis Calder Center site since the early 1970s. There was an increase in human population density in the town of North Castle, NY in Westchester County from 9,591 (1970) to 11,841 (2010) according to the Decennial Census 1950–2010 (United States Census Bureau, 2010). An increase in population density often results in construction of more housing and roads and subsequent fragmentation of forests, which could alter current species assemblages. We identified the most drastic change in relative abundance over time for three species; Ni. tomentosus, O. noveboracense and Necrophila americana (Fig. 4). In contrast, relative abundance of Ni. tomentosus increased, which could be due to the fact that this species is an ecological generalist that becomes more common after habitat degradation. There was a substantial decrease in O. noveboracense since 1974. Since univoltine arthropod species are more affected by habitat loss, (Kotze et al., 2011) the reduction in O. noveboracense may be due to their inability to effectively compete with multivoltine species, as well as their limited dispersal ability in warmer temperatures (Ratcliffe, 1996). Lastly, the most drastic decline was in Necrophila americana, which is perhaps due to its large body size (smaller bodied generalists can survive in more disturbed/urban habitat; Gibbs & Stanton, 2001; Elek & Lövei, 2007) and its preference for field habitat (Ratcliffe, 1996), which may be more limited in the area in 2015 due to fragmentation and the reduction in agriculture. Two previously-observed species, Necrodes surinamensis and Necrophilus pettiti, were completely absent in our contemporary sample. Necrodes surinamensis is nocturnal and highly attracted to artificial lights, causing Ratcliffe (1996) to state concern for this species in cities where increased nocturnal lighting is common. This may be the reason for the decrease or even the extirpation of this species at this site. As for Necrophilus pettiti, it is a flightless carrion beetle species (Peck, 1981), possibly limiting its dispersal and survivability in increasingly urbanized areas. Alternatively, our use of hanging traps rather than pitfall traps as in Pirone & Sullivan (1980) may explain the absence of this flightless species in our study.

In their study, Gibbs & Stanton (2001) discuss several ecological factors that may influence the presence and relative abundance of carrion beetles in urban areas. The first limiting factor is the availability of carcasses. However, urbanization can lead to an increase in abundance of some birds and small mammals (Faeth, Bang & Saari, 2011; Pickett et al., 2011). Forests in New York City typically contain large rodent populations (particularly white-footed mice and chipmunks), as well as abundant songbird populations (Ekernas & Mertes, 2006; Seewagen & Slayton, 2008). Availability of bird and small mammal carcasses in urban parks may be higher than previously appreciated, and may explain the abundance of some generalist species (Ni. tomentosus) in NYC.

Competition with scavengers may also limit carrion beetle abundance and diversity. Competitors for carrion include other invertebrates (flies and mites—Gibbs & Stanton, 2001) and many vertebrate mesopredators that are abundant in cities (raccoons, opossums, coyotes, skunks—DeVault et al., 2011). However, Sugiura et al. (2013) recently reported that resource competition between invertebrates and vertebrates for carrion was less prevalent than previously thought. Vertebrate competitors are likely present at all of our study sites but may not substantially influence carrion beetle abundance, particularly if they are utilizing food resources provided by humans in cities. As for intraspecific competition with other carrion beetle species, Scott (1998) found there to be competition between Ni. orbicollis and Ni. defodiens based on temperature. Trumbo & Bloch (2002) found that Ni. defodiens can locate carcasses sooner than other species, but Ni. orbicollis uses cues from Ni. defodiens to locate and subsequently dominate carrion. We identified a higher abundance of Ni. orbicollis than Ni. defodiens, especially in urban and suburban sites (Fig. 2). Greater abundance of Ni. orbicollis in our sites could be due to these competitive abilities that aid Ni. orbicollis in locating and dominating prey more effectively. Lastly, intraspecific invertebrate competitors such as flies are often very prevalent in human altered landscapes (Kavazos & Wallman, 2012). Flies are known to quickly locate carcasses and may outcompete beetles on carrion (Scott, 1998; Scott & Traniello, 1990; Trumbo, 1990; Gibbs & Stanton, 2001). If there is high fly abundance in NYC parks, our study indicates that this intraspecific competition is not restricting beetle abundance in urban forests compared to suburban and rural forests. Some carrion beetles have also evolved adaptations to thwart fly competition. For example, Ni. tomentosus was the most abundant carrion beetle in urban forests in this study, and may be thriving partially due to its cooperative burying behavior that rapidly conceals carcasses from flies during times of day when flies are most active (Scott, Traniello & Fetherston, 1987).

Gibbs & Stanton (2001) also identify soil compaction as a negative influence on carrion beetles in urban areas. Soil compaction is characteristic of urban forests due to trampling by humans and other factors, and may impede the ability of beetles to dig and bury carcasses in urban soils (Gibbs & Stanton, 2001; Pouyat et al., 2007). The relatively shallow burying of Ni. tomentosus may be another factor promoting the success of this species in NYC (Fig. 2). In the study by Wolf & Gibbs (2004) they found soil compaction to not be correlated with urbanized land, but was partially correlated with forest extent. Additionally, larger forest area has been shown to support greater diversity of burying beetles (Gibbs & Stanton, 2001). NYC’s urban parks seem to contain sufficiently extensive tracts of forest for carrion beetles to persist despite possible soil compaction.

We found no trend in body size (as classified via Scott, 1998; Gibbs & Stanton, 2001), across habitat specialization (from Gibbs & Stanton, 2001), or with continuous forest habitat area compared to abundance of species found in urban, suburban and rural sites. Our results of body size and habitat specialization were again contradictory to past burying beetle literature where Ulrich, Komosiński & Zalewski (2008) found a negative association between body size of necrophagous beetles and distance from the city center in northern Poland. Gibbs & Stanton (2001) also reported that carrion beetles thriving in urban areas were often small-bodied and habitat generalists. Magura, Tóthmérész & Lövei (2006) used even more stringent statistical measures to assess carabid beetle body size across an urbanization gradient. These studies measured body length and/or biomass of each individual, whereas our study was limited by gross approximation of body size classes, which may have caused us to miss this effect. In a broader analysis of 69 beetle species (Davies & Margules, 2000), body size was not correlated with fragmentation. The authors argued that the relationship between extinction risk and body size is very complex and influenced by other factors like spatial scale, population fluctuation, and longevity.

In general, our study shows that within NYC, urban parks are able to house nearly the same community as continuous forest tracts in the rural surrounding areas despite local impervious surface and ecological/life history variability across species. These results highlight the importance of maintaining and conserving large areas of forest throughout NYC within city parks commonly used for human recreation.

Conclusions

Urban parks have the potential to house diverse habitats rich in biodiversity (Kotze et al., 2011) of both plants and animals (Angold et al., 2006). Even highly modified landscapes containing small reserves comprising ample green space have the potential to house large beetle diversity (Watts & Lariviere, 2004) and high abundance of other arthropod species (Bolger et al., 2008; Faeth, Bang & Saari, 2011). The maintenance of arthropod biodiversity in urban parks may ultimately be mediated by human influence on plant communities (Faeth, Bang & Saari, 2011). Alternatively, arthropod species thriving in urban habitats may be preadapted for tolerance to fragmentation and high colonization potential (Sattler et al., 2010). A current review on biodiversity in cities suggests that patch area and corridors have the strongest positive effect on biodiversity and that we need to maintain sites with larger than 50 hectares to prevent rapid loss of sensitive species (Beninde, Veith & Hochkirch, 2015). Maintenance of carrion beetle diversity in NYC will stabilize the interconnectedness of urban food webs, aid in nutrient cycling, and promote natural decomposition of carcasses (Beasley, Olson & Devault, 2015) in our urban parks. Sustaining the ecosystem services provided by carrion beetles will require conservation of large, continuous forest tracts in urban parks. Greater connectivity between small green areas, and connectivity between the urban core and surrounding forested areas will promote the biodiversity potential of small patches (Do & Joo, 2013). A “land sparing city” approach is one way to maintain essential ecosystem services (Stott et al., 2015) provided by carrion beetles in the New York City area.

Supplemental Information

Table S1 Hill’s True Diversity numbers; q = 0, q = 1 and q = 2 calculated for comparison of species diversity indices.

Equations retrieved from (Jost, 2006).

Click here for additional data file.

Table S2 Relative abundance (mean) across different site classification (urban, suburban, rural) comparing beetle size classes (small, medium, large) and habitat specialization (generalist, specialist; classifications from Gibbs & Stanton, 2001.

Click here for additional data file.

Figure S1 Relationship between carrion beetle (A) species richness and (B) species diversity (1/D) with percent forest area at all sampled sites (site abbreviations found in Table 1 and Fig. 1).

Click here for additional data file.

Data S1 Raw data on carrion beetles collected at 13 sites along an urban-to-rural gradient

This file includes a single entry for every individual carrion beetle collected during this study, including information on species, site (codes follow Table 1 in the manuscript), site coordinates, and Start and End dates of the surveys in which the beetle was collected.

Click here for additional data file.

We thank Matthew Combs and Jane Park for photographing carrion beetles, and Elizabeth Carlen, Carol Henger, and Emily Puckett for helpful comments on the manuscript. Gabor Lovei and two anonymous reviewers also provided very thorough and constructive suggestions for improving the manuscripts.

Additional Information and Declarations

Competing Interests

Author Contributions

Field Study Permissions

Data Availability

The authors declare there are no competing interests.

Nicole A. Fusco and Anthony Zhao conceived and designed the experiments, performed the experiments, analyzed the data, wrote the paper, prepared figures and/or tables, reviewed drafts of the paper.

Jason Munshi-South conceived and designed the experiments, performed the experiments, contributed reagents/materials/analysis tools, wrote the paper, reviewed drafts of the paper.

The following information was supplied relating to field study approvals (i.e., approving body and any reference numbers):

Permission to collect carrion beetles was granted by the New York City Department of Parks and Recreation, the Rockefeller State Park Preserve, and the Connecticut Department of Energy and Environmental Protection (Permit No. 1214008).

The following information was supplied regarding data availability:

The raw data has been supplied as a Supplementary File.

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
