# Peer review of "Urban forests sustain diverse carrion beetle assemblages in the New York City metropolitan area"

_PeerJ, doi:10.7717/peerj.3088_

## Round 0.1 · original submission · Major Revisions

The three reviews are all thorough and offer numerous suggestions for improvement. I won't list them all here, but I think going through the reviews, line by line, and responding will lead to a much improved and more clear paper. Let me know if you have any questions.

Reviewer 1 ·

Basic reporting

Not all raw data are available

Experimental design

Statistical analyses should be expanded/revised (see comments below)

Validity of the findings

No Comments

Additional comments

Major suggestions
1) I think that the authors, in addition to Simposn's index, should use some other index of diversity/dominance/equitability to investigate spatial and temporal variations in community structure.
2) Table 3 is not easy to read. Why do not use a cluster analysis?
3) In particular, I would suggest to better explore patters of beta-diversity. For this, the authors might use, instead of Jaccard's index alone, a beta-diversity partitioning approach. I suggest to use the approach of Baselga et al. (2007) and Baselga (2010, 2011) for partitioning the overall ß diversity (ßsor) into true species replacement or pure turnover (ßsim) and nestedness (ßnest) components.
4) I suggest to replace the bar plots of Figure 2 with box plots.
5) Figure 3 should be deleted, since the analyses are not significant.
6) Table 2. Adjust species names. Report abundance data, not only presence

Minor corrections
lines 138 & 166: student's → Student's
line 158: rto → to
line 171: .. → .
line 174: delete comma after species
line 178: delete comma after sites
Avoid citing Sokal and Rohlf 2011 for well known tests

·

Basic reporting

An interesting study, but the analysis and reporting has some serious faults. General comments below contain points of logic, execution, reporting and language.

Experimental design

Restricted sampling, needs justification. See under general comments below. Some more detail needs reporting. Your sampling would be limited even with three one-week sampling session, but most of the analysis is based on one period - i.e. one week of sampling, which is not acceptable.

Validity of the findings

The detailed comments indicate where the validity, in my view, is overextended. You will have to address these. See below

Additional comments

15, 17 – no need to indicate the name of the higher category. move the "Coleoptera" to the seconded mention.
21 reference to your own research should be in past tense
23 Gradient: the MS phrases this as an "urban-rural gradient" but this is incorrect. The base of comparison is the rural area, and the main question is whether urbanisation impoverished the original diversity of the rural area, or what is retained in urban forests from the original fauna. Therefore, it is more correct to present this as a "rural-urban gradient". Modify throughout. Rephrase accordingly.
24 I would not call it a more in-depth study. It is only a comparison with an earlier sampling. Useful, but not with a larger efforts, etc. So do not inflate the importance of this element. Write "comparative" Move it later, because here, after the mention "more in depth", reader expects some more detail about this – and the following sentence jumps back to the main topic, the gradient.
29 shallow – if "shallower", compared to what?
32 rephrase. Clumsy and ambiguous. What is "10 carrion beetle relative abundance"?
44-51 this is a little careless. It basically states that there is a hodgepodge of data and no generalisation is possible. The situation is different. For example, ground beetles do not become more species poor in urban areas but forest specialists suffer (see review in Magura et al. 2010 Global Ecol Biogeogr). Revise this section because the summing up that "Given the extreme variety of life history traits and habitat use among arthropods, responses to urbanization may be difficult to predict." is a poor summary of available knowledge.

58-60 due to resource concentration, esp. an abundance of human food waste, urban areas usually support a high abundance of small mammals, mainly mice and rats. It is not correct to state that urbanisation causes "reduced abundance of small mammals"
End of introduction: carrion beetles can be supported because of their special need of carrion. This may be available, because urban areas have lots of small mammals (and they possible can utilise waste, too), thus their main food source can be superabundant. If there is a rich carrion beetle diversity, for the above reasons, it cannot be generalised that "urban forests in NYC are currently undervalued as reservoirs of arthropod biodiversity." Rephrase.

The traps were then in the air? If so, beetles could only approach the traps by flying. It is more usual to trap carrion beetles in pitfall traps. Give a reason why you did not do this, and indicate whether all potential species (i.e. the species present in the rural areas) are all able to fly. Not all Silphidae are able to fly. The trapping period is very short with respect to the growing season/activity season. Give a justification why such a short trapping gives reliable results?
128 this decision is doubtful. so you considered all the material collected as constituting 100%? This cannot be justified: the distances were too large, so that you did not sample the same assemblages. therefore it is better to consider locations as independent samples, each with its own structure (which can be expressed by relative abundance). Then you will have mean relative abundances per urbanisation stage, with variability, and these can/should be compared.
131-132 you have to justify this decision with something. Simply stating it is not acceptable. Also, why was the main analysis restricted to the last one week long (!) sampling? This is not acceptable in any ecological study.
135 more needs to be written about this, at least how did you treat the historical study, and how big was it?
138 Student's (should be capitalised)
142 what was your definition of species richness? This is not clear, especially as your sampling is so superficial.
165 now you use mean relative abundance – but it is far from obvious how did you calculate this. I suggest you always use this parameter only, and define clearly.
169 citation wrongly placed. Sokal & Rohlf say nothing about the forest extent – they may mention the statistical method 8general linear regression) – is so, place the citation there. I also have problems with the independent variable, see comments to S4 below.
174 do not capitalise silphid
175 ff. this is a usual comparison and as such OK but not clear what do you mean "mean relative abundance of all species across urban, suburban, and rural site classes". Rephrase to clarify. you can use abundance – which was calculated as relative abundance – repeated mention only creates confusion.
180 with one species so dominant, I suggest it merits a separate analysis at the individual species level for this one. Add a short section.
189 ff this is best put under the subsequent heading
190 summer doesn't need to be capitalised (multiple)
more details needed about the historical collection 8ealier) – compare if the abundance has changed, or species richness? Right now it is bit mixed up.
201 have consistency – previously you mentioned species richness first, then diversity.
204-208 "the" is missing form a number of places.
209 delete rest of sentence from indices – not the info in ()s
There is not a very high species richness, so your parameter space is not vast. This is why you can have complete identity (index at maximum). Discuss this aspect later. You certainly are not dealing with a high level of diversity here – so from this it is a gross exaggeration to conclude that all invertebrate biodiversity may be saved by NYC urban forest patches. It could be useful to compare the species richness between this group and for example, carabids that are reasonably well studied in rural-urban gradients world-wide (see Magura et al. 2010 Global Ecol Biogeog review). Discussing this would put some proportion into your paper.
219 etc. – again, this may be simply due to the restricted range of species possibly present
243 not sure you can generalise like this. In many studies, species richness has decreased from rural to urban, and when dealing with an urbanisation gradient starting from a forested rural habitat, specialists decreased, small species became more prevalent, species in urban areas have higher dispersal power. Overall, your statements are generalisations that cannot be sustained as stated.
249 Magura et al. 2010 is not a formal meta-analysis. It is an evaluation of several hypotheses based on published data collected using a uniform experimental setup (Globenet) but not a meta-analysis.
304 I dou7bt that Kotze et al. 2011 discussed carrion beetles, and esp. not O. novaeboracense – thus the citation is misleading. Delete or rephrase.
306-307 this was also found in ground beetles in Denmark by Elek & Lovei 2007, Acta Oecologica
313 Ne. pettiti – how could you collect a flightless species in your traps when they hung in the air?
326 no need for Italics for these animal names
340 in order this to be effective, one has to prove that carrion with fly eggs is unusable for carrion beetles. Was this shown/proven?
All in all, this section is a little extensive – it seems most of the factors mentioned here are not important or limiting for carrion beetles. Shorten this part.
357 your body size was crudely approximated, only three classes. No surprise that you did not find much difference. Admit and discuss the limitations. Others (e.g. Ulrich et al. 2008, in Poland) studied this aspects in more detail and their findings are more reliable. Your results cannot counter those of Ulrich & al., and the reason for not finding differences can be due to this methodological difference.
364 the study need to be cited also here. See also Magura et al. 2006 Basic Appl Ecol for a counterexample
367 no, as stated above, your methods may not be sensitive enough



Check the reference section arrangement – e.g. Stott et al. 2015 is not in the right place.

Table 4 is not needed – the information can easily be presented in a more economic way in the text.
I am not sure if fig 1 is informative or meaningful. It is too small scale to see much of the suburban-urban sites, and e.g. on the basis of this map, the SWP site is misclassified as suburban – it looks very different from the other three suburban sites.
fig 2A – vertical axis has no unit (it is probably %). This type of figure is difficult to interpret, so I suggest to use an alternative,
Fig 2B – not clear what is the intent here, and why is it not a figure of its own right? It does not seem to be related to the A part of the same figure. Needless frames on both A & B.
Fig 2C is also not too informative. The figure overall is too multi-coloured, which is not informative.
3A not clear what species richness is here; a statistical approach would be better, i.e. the mean no. of species/trap. Symbols are too small. this figure also breaks the principle that a figure has to be understandable without reference to text or the other figures. Frame around figure unnecessary.
Fig 4 – the data points should be bigger. There is no need to write out the colour codes in text - they are on the figure itself already.
Supplementary S1 – not clear how total relative abundance is defined here, nor is it intuitive. Choose another parameter, such as no. of individuals / trap or similar. The fat boxes are also ugly – and the width of the boxes carries no information. Consider using the Tufte plot instead.
S2 – I suggest this should be arranged as a rank-abundance figure, i.e. the highest abundance first, etc.?
Table S1 – not sure I trust the ANOVA results. There is too little difference in the F values between medium and large beetles, the degrees of freedom will be identical, yet the p value is very different. Are you sure this is correct? no need for the many horizontal lines.
Table S1 & S2 can easily be combined
S3 & S4 – the extent of continuous forest area (how defined?) is not necessarily relevant – above a certain extent, beetles will start forming independent populations, which were not sampled anyway. Find a different, better independent parameter, like the % of forest area within a X km radius?

Reviewer 3 ·

Basic reporting

This article meets the criteria for basic reporting standards.

Experimental design

The experimental design is sufficient to address the questions asked in this manuscript.

Validity of the findings

The statistical models are generally appropriate (see general comments for details).

Additional comments

Much of the literature on biodiversity responses to urbanization tend to assume reductions in diversity; this study shows that carrion beetle diversity is unaltered by urban development, and because this work challenges the assumption of biodiversity loss, this study makes an important contribution to the literature. Overall the study is well conducted, analyzed and written (though a heavy edit on typos is needed). I have a few suggestions to consider for improving the clarity of this manuscript.
Line 132 – what is the justification for only using the final census point for this analysis?
Line 136 – are the timeframes for the historical samples known? i.e. do these fall within the range of months over which the contemporary samples occurred?
Line 143 – Simpson’s index doesn’t always do well with rare species. Have you confirmed that alternative diversity metrics (Chao, Hill indices) yield similar results? Or possibly the beetle species are all fairly abundant in your samples?
Line 151 – at first, the lack of spatial autocorrelation being accounted for in these GLM’s concerned me, but there is only a single index (richness, diversity) for each site in this analysis, right?
Line 152 – for the GLM’s – did the richness data follow a normal distribution? Often these are Poisson distributed since these are counts.
Line 158 – typo “rto obustly”
Line 163 – if there are enough data to bin these into size classes, why not use the body size data directly as a continuous covariate?
Line 179 – test statistics?
Line 288 – not sure what this means “for = wetlands”

---

## Round 0.2 · accepted · Accept

I really like it when the authors seriously consider the improvements suggested by the reviewers. This is how the publication process is supposed to work. Thanks.